# Photothermal Effects of High-Intensity Laser Therapy on the Superficial Digital Flexor Tendon Area in Clinically Healthy Racehorses

**DOI:** 10.3390/ani12101253

**Published:** 2022-05-12

**Authors:** Paulina Zielińska, Maria Soroko, Maria Godlewska, Karolina Śniegucka, Krzysztof Dudek, Kevin Howell

**Affiliations:** 1Department of Surgery, Wroclaw University of Environmental and Life Sciences, pl. Grunwaldzki 51, 50-366 Wroclaw, Poland; paulina.zielinska@upwr.edu.pl (P.Z.); godlewskamaria23@gmail.com (M.G.); sniegucka.karolina@gmail.com (K.Ś.); 2Institute of Animal Breeding, Wroclaw University of Environmental and Life Sciences, Chelmonskiego 38C, 51-160 Wroclaw, Poland; 3Faculty of Mechanical Engineering, Wroclaw University of Technology, Lukasiewicza 7/9, 50-231 Wroclaw, Poland; krzysztof.dudek@pwr.edu.pl; 4Microvascular Diagnostics, Institute of Immunity and Transplantation, Royal Free Hospital, Pond Street, London NW3 2QG, UK; k.howell@ucl.ac.uk

**Keywords:** thermography, high-intensity laser therapy, racehorse

## Abstract

**Simple Summary:**

High-intensity laser therapy (HILT) is becoming an increasingly popular form of physiotherapy for tendon injuries. This short communication paper discusses the photothermal effects of HILT on the superficial digital flexor tendon of the hindlimb in clinically healthy racehorses. A thermographic assessment of the surface temperature change in the superficial digital flexor tendon was performed before and immediately after high-intensity laser therapy. The results indicate a statistically significant increase after treatment. These findings may be helpful in determining the optimal HILT parameters needed for treating tendon injuries.

**Abstract:**

The aim of the present study was to assess the photothermal effects of high-intensity laser therapy (HILT) on the superficial digital flexor tendon (SDFT) of the hindlimb in racehorses. It was conducted on 18 clinically healthy thoroughbreds that were subjected to thermographic examination to measure surface temperature changes in the SDFT. This was performed before and immediately after HILT. This revealed statistically significant differences in the temperature of the skin surface overlying the flexor tendons (*p* < 0.001). The surface temperature of the area examined was higher by an average of 3.5 °C after HILT, compared with the temperature measured before HILT. Our results prove that HILT has a photothermal effect in treating soft tissue. This finding can be helpful in determining the appropriate parameters for monitoring the short- and long-term effects of HILT.

## 1. Introduction

High-intensity laser therapy (HILT) has recently been introduced as a new treatment option in the field of equine veterinary medicine and rehabilitation [1,2]. HILT employs high-power (class IV) laser-generated focused light at various wavelengths within the visible to the infrared part of the electromagnetic spectrum (790–1100 nm) [3,4]. HILT delivers high-intensity light energy to deep tissues in short pulses (120–200 µs) [5]. It causes minor and slow light absorption by the chromophores, which increases mitochondrial bioenergetics by causing extra adenosine triphosphate synthesis and oxygen consumption, as well as RNA and DNA production, resulting in tissue stimulation [6]. Various studies have shown that laser light has anti-inflammatory and analgesic effects and that it reduces edema [7,8]. HILT also has photothermal properties, resulting from the transformation of high-energy light into heat in tissue [9,10]. Increasing skin temperature affects blood flow, vascular permeability, and cell metabolism through vasodilatation. Heat also increases oxygen uptake, the activity of destructive enzymes such as collagenase, and the catabolic rate, while accelerating tissue healing [11,12]. Using a regular waveform, with a peak power of up to 3 kW and an extended time between pulses (to decrease thermal accumulation phenomena), HILT is able to rapidly induce photothermal effects in the target tissue with very low histological risk. This occurs through the provision of high energy density to ensure that more photonic energy effectively penetrates and reaches the deeper target tissues while preventing the skin from overheating, resulting in the desired high level of energy in a short time period [13].

HILT is regularly used in equine veterinary medicine to treat tendon and ligament injuries [14]. Pluim et al. [15], who studied 150 sport horses treated with HILT for tendinopathy and desmopathy, found a statistically significant overall improvement in both lameness and ultrasonographic score the day after a two-week laser therapy period, as well as four weeks later, across all injury types and stages. The main limitation of their study was the lack of a control group; however, the results offer a promising prognosis for chronic tendon injuries. Quiney et al. [16] reported on two clinical cases of horses with a primary injury of the carpal medial collateral ligament. Both of the horses demonstrated ultrasonographic improvement after HILT treatment. Finally, in our previous study involving 29 cases of flexor tendon and suspensory ligament injuries, HILT was found to have significantly reduced the size of the injury in a laser-treated group, compared with a laser-sham group [17]. 

Although numerous HILT experiences have been reported in the literature, thus far they have only been based on clinical cases and outcomes described by practitioners. However, more research needs to be conducted to examine the photothermal effects of HILT on healthy tendon tissues in order to more accurately determine the correct parameters for tendon injury treatment. In our previous study, infrared thermography was used to evaluate the effects of HILT on the body surface temperature overlying the tarsal joint in clinically healthy racehorses. Following treatment, body surface temperature was found to have increased by approximately 2.5 °C. This study was the first to identify the photothermal effects of HILT on a joint [18]. 

Previous studies have not investigated the photothermal effects of HILT on the body surface temperature in clinically healthy soft tissue. Therefore, the aim of the present study was to assess the photothermal effects of HILT on the superficial digital flexor tendon (SDFT) in the hindlimb of clinically healthy racehorses. The hypothesis was that HILT would increase the body surface temperature in the clinically healthy tendon area. 

## 2. Materials and Methods

### 2.1. Animals and Study Design

Eighteen clinically healthy 3–4-year-old thoroughbred racehorses, in regular training with the same trainer, were studied at Partynice Race Course in Wroclaw, Poland. The horses were housed in individual stalls with common management at the same stable. The study was approved by the Local Ethical Committee for Experiments on Animals (Wroclaw, Poland; registration number 003/2020).

On the day of examination, each horse (at rest, exercised daily), was subjected to thermographic examination in order to determine body surface temperature changes in the plantar surface of the SDFT, which is in the middle one-third of the third metatarsal bone of the left hindlimb. This was performed just before and immediately after HILT application. None of the horses had white marks in the treatment area and skin color was assessed as being pigmented. 

The treatment area was irradiated with a class IV Polaris HPS laser (Astar, Bielsko-Biała, Poland), with two wavelengths delivered simultaneously: an 808 nm (an AlGaAs laser with a maximum output power of 8 W) and a 980 nm (an InGaAs/AlGaAs laser with maximum output power of 10 W). Different treatment parameters were used for each wavelength to avoid uncontrolled thermal effects, tissue destruction, and skin burns (Table 1). HILT was carried out on weight-bearing limbs using a manual scan and contact technique by the same veterinarian, who was experienced in laser therapy. The contact technique was used to prevent the laser light from scattering and was performed without any pressure on the tissue, with minimal probe contact on the horses’ coats. Scanning was performed in both the transverse and longitudinal directions of the limb. The laser handpiece was held perpendicular to the skin surface with a treatment area of 22 cm^2^. 

The protocol for the thermographic examination was similar to that of our previous studies [18,19,20]. Images were taken in the stable corridor, within an enclosed stable, to avoid the effects of air drafts and exposure to sunlight, at a constant ambient temperature of approximately 15 °C. They were taken at a distance of approximately 1.5 m from the animal using a VarioCam hr Resolution infrared camera (uncooled microbolometer focal plane array, resolution 640 × 480 pixels, spectral range 7.5–14 mm, InfraTec, Dresden, Germany). 

The thermographic images of the plantar surface of the SDFT were taken just before and immediately after HILT treatment. The emissivity (ε) was set to 1 for all readings [21]. The average surface temperature of a rectangular area, superimposed over the area treated (Figure 1), was calculated using IRBIS 3 Professional software (InfraTec, Dresden, Germany).

### 2.2. Statistical Analysis

STATISTICA v. 13.3 (TIBCO Software Inc. Palo Alto, CA, USA) was used for the statistical analysis of the results. The distributions of the temperatures measured were checked for normality using the Shapiro–Wilk test (*p* = 0.213 before; *p* = 0.705 after). The empirical temperature distribution, measured before and after treatment, did not differ significantly from a normal distribution. Parametric tests were used in further analyses. The critical level of significance was set at *p* < 0.05. For the quantitative variables, the following basic descriptive statistics were estimated—mean values (M), standard deviations (SD), medians (Me), lower (Q1), and upper (Q3) quartiles—as well as the extreme values—lowest (Min) and highest (Max). The significance of the differences in the mean values in the two dependent groups (horses before and after treatment) was calculated using Student’s t-test for dependent variables.

## 3. Results

The average body surface temperature in the SDFT area increased significantly after HILT (*p* < 0.001). The body surface temperature of the area examined was higher (by a mean of 3.5 °C) after HILT, compared with the temperature before HILT (Table 2). The temperature change was significantly different from zero (*p* < 0.001).

## 4. Discussion

This study has demonstrated the photothermal effect of HILT, when applied to healthy soft tissue. The body surface temperature overlying the SDFT increased by an average of 3.5 °C, confirming the hypothesis of the study. In our previous research, we reported a significant increase in both body surface temperature and the diameter of the cranial branch of the medial saphenous vein at clinically healthy tarsal joints after HILT [18]. Similar results were found regarding the fetlock joint, where the temperature after HILT increased by 3 °C [22]. A study on defocused CO_2_ laser therapy has reported a significant increase in the temperature of the skin and subcutaneous tissue in the fetlock joint area [23].

Notable results were also found in another of our previous studies, where we reported on differences in the effects of HILT on pigmented and non-pigmented skin, at a clinically healthy fetlock joint. We found an increase in the body surface temperature of horses with pigmented skin and a decrease in horses with non-pigmented skin. Although the vein diameter in both groups increased after HILT, the differences between groups were statistically insignificant [24]. 

The thermal effects of laser therapy measured on skin surfaces are the result of laser light absorption [6]. When laser therapy is performed using the contact method, heat energy can be transferred from the laser probe to the skin by conduction. In their study, Joensen et al. [25] used thermography to assess skin temperature after laser therapy using a stationary contact method. They found that the laser probe was not responsible for elevations in skin temperature. However, if therapy is performed using a labile contact method, the friction created by the movement of the probe over the treatment area can increase body surface temperature. To the best of our knowledge, there have not been any studies assessing the thermal effects of probe friction during laser therapy. The lack of a control group for assessing the possible thermal effects resulting from conduction or friction was the main limitation of the above-mentioned study.

Tendon lesions are common in equine athletes [26,27], with slow and incomplete repair mechanisms resulting in a high chance of re-injury [28,29,30]. One of the reasons for limited tendon tissue healing capacity is relatively poor vascularization [31,32]. Adequate blood supply is necessary for the transport of nutrients, inflammatory mediators, and proteolytic enzymes in and out of a tendon [33].

Based on the previously reported beneficial influence of HILT on equine tendon injury repair and the results of our own studies, it is plausible to assume that some photothermal energy may be transferred into deeper tissue during the application of HILT over the SDFT. It is known that the photochemical and photothermal effects of HILT can increase blood flow, vascular permeability, and cell metabolism, and thus help stimulate collagen production within tendons, thereby repairing injuries [34]. Hong et al. [35] have reported that the effects of HILT can promote fibroblast proliferation, and thus result in significant increases in collagen synthesis, granulation tissue formation, and extracellular matrix production.

The final effects of HILT on treated tissue depend on several factors, such as wavelength, irradiation mode, pulse duration, pulse time interval, energy fluence, power output, and irradiance. The HILT treatment parameters used in the present study did not contribute to any clinical abnormalities, and no adverse effects were observed in the irradiated limbs or the horses’ behavior. As such, it can be concluded that the HILT parameters used are safe and well tolerated by horses.

## 5. Conclusions

Our results have demonstrated the photothermal effects of HILT on soft tissue. They can be helpful in determining appropriate and safe HILT parameters for the treatment of SDFT injuries and inflammation. Our study only involved 18 horses, which may have limited the statistical power of the results. The small sample size and limited area of treatment make it impossible to draw conclusions about the actual photothermal impacts of HILT on other types of non-pathological soft tissue. Additional research is required to determine the influence of HILT on different types of tissue in specific parts of the body, and on pigmented and non-pigmented skin.

## Figures and Tables

**Figure 1 animals-12-01253-f001:**
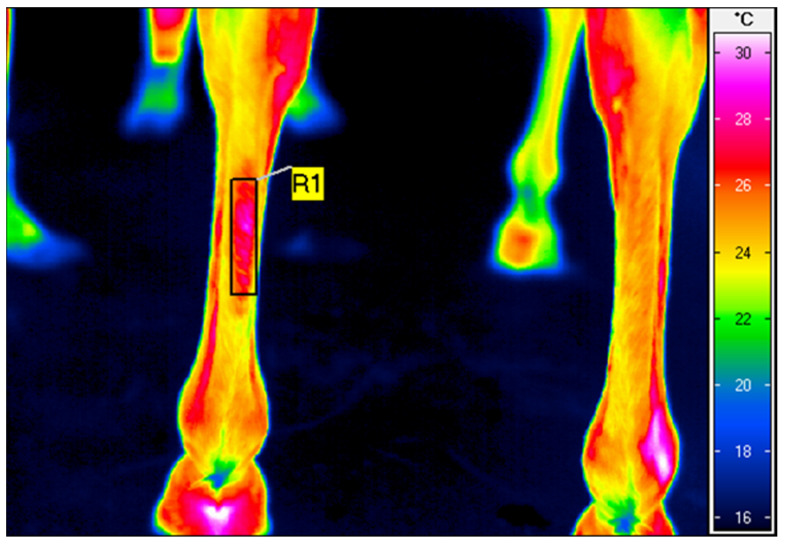
A thermographic image of the plantar aspect of the middle one-third of the third metatarsal bone of the left hindlimb in the superficial flexor tendon region, taken after high-intensity laser therapy application. The rectangular area (R1) indicates an average body surface temperature of 26.3 °C.

**Table 1 animals-12-01253-t001:** High-intensity laser therapy parameters for treatment of the superficial digital flexor tendon of the hindlimb.

Wavelength (nm)	Type of Diode Laser	Power Output (W)	Energy Density (J/cm^2^)	Frequency (Hz)	Time of Irradiation (s)
808	AlGaAs	4.0	25	700	171
980	InGaAs/AlGaAs	4.0	16	1000	125

**Table 2 animals-12-01253-t002:** Significance of differences in mean temperatures before and after high-intensity laser therapy.

	Before HILT N = 18	After HILT N = 18	Change	*p*-Value
T_avg_ (°C)				<0.001
M ± SD	21.2 ± 3.0	24.6 ± 2.9	3.5 ± 2.91	
Me (Q1; Q3)	21.3 (18.4; 23.4)	24.4 (22.4; 26.6)	3.4 (2.5; 4.5)	
Min–Max	16.7–26.5	19.9–29.1	0.9–7.4	

Abbreviations: HILT—high-intensity laser therapy; SD—standard deviation.

## Data Availability

The data that support the findings of this study are available from the corresponding author (M.S.), upon reasonable request.

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
