# Peer review of "Photothermal Effects of High-Intensity Laser Therapy on the Superficial Digital Flexor Tendon Area in Clinically Healthy Racehorses"

_animals, 2022, doi:10.3390/ani12101253_

Round 1

Reviewer 1 Report

Thank you for clarifying the points I raised and revising the manuscript.

Reviewer 2 Report

I am affraid but authors did not follow my suggestion:

Laser is said to act through photo-biomodulation, that involves essentially a series of tissue responses to laser: stabilization of the cellular membrane, enhancement of ATP production, (not synthesis) vasodilatation, acceleration of leukocytic activity, increased prostaglandin synthesis, reduction in interleukin 1, enhanced lymphocyte response, increased angiogenesis, temperature modulation, enhanced superoxide dismutase levels, decreased C-reactive protein and neopterin levels. All these tissue activities reduce acute inflammatory response. Photo-biomodulation also produces analgesia by increasing beta-endorphins and nitric oxide production, decreasing bradykinin levels, ion channel normalization, blocking depolarization of C-fiber afferent nerves, increasing nerve cell action potentials and release of acetylcholine, and axonal sprouting and nerve cell regeneration. The other biological response includes DNA and RNA synthesis, an increase in cAMP levels, protein and collagen synthesis and cellular proliferation. 

Furtheremore, the hypotesis is little bit simply, :

https://pubmed.ncbi.nlm.nih.gov/15315730/

Reviewer 3 Report

Thank you to the authors for their modifications to the manuscript, which is significantly improved in its current version. The authors have satisfactorily responded to each of this reviewer's comments.

This manuscript is a resubmission of an earlier submission. The following is a list of the peer review reports and author responses from that submission.

Round 1

Reviewer 1 Report

Thank you for submitting your study and presenting your results of HILT to the SDFT region in horses.

I have comments that I would like to be addressed before considering that this paper suitable for publication.  

You main focus is on skin surface temperature change and this being a result of the HILT.  You briefly mention absorption of heat into deeper tissues but this is an area that needs expansion.  Why do you consider this heat  transference as beneficial, and is it beneficial for every stage of SDFT lesions? If you are associating skin surface changes with therapeutic effects, does this need to be delivered by HILT? Could moist heat or diathermy/radiofrequency have the same effect?

Apart from your own previous study, you have not discussed the previous research on pigmented/non-pigmented skin and laser absorption. (for example see - Duesterdieck-Zellmer, K.F., Larson, M.K., Plant, T.K., Sundholm-Tepper, A. and Payton, M.E., 2016. Ex vivo penetration of low-level laser light through equine skin and flexor tendons. American journal of veterinary research77(9), pp.991-999.) You do not report on the pigmentation of the skins in the horses used in this study. Where they all the same?

Is the reduction in skin temperature rise potentially a sign of absorption in deeper tissues, rather than more superficially?

I also have ethical concerns as you do not report on whether the horses were observed for discomfort or pain during the application of the laser. This is particularly relevant because you do state that ' Different treatment parameters were used for each wavelength to avoid uncontrolled thermal effect and tissue destruction or skin burns.' Please can you expand on any behavioural observations made

Reviewer 2 Report

Dear authors, 

Overall, The manuscript is well written, some small modifications are needed. 

Ln 42-43: This sentence is awkward. Please introduce more science behind mitochondrial respiration and the impact of HILT on these processes.

Reviewer 3 Report

Thank you for the opportunity to review this short communication. The manuscript is well written and easy to follow. Expansion on the relevance and clinical applicability as well as next steps is recommended. See specific comments below.

19 – temperature not temperatures

 21 – increase in surface temperature

22 – unclear what word is intended with opportunely – perhaps substitute for optimal

23 – revise to ‘treatment of tendon injuries’ vs ‘tendon injuries treatment’

25-26 – do not need to repeat that the horses are clinically healthy just say once

43 – and not or RNA or DNA?, revise sentence to add ‘and’ before resulting

45 – perhaps revise to ‘reduces edema or swelling’ rather than ‘antiedematous’

49 – could you expand briefly on lack of histological risk? Cite studies if they exist

54-55 – is this indicating that ultrasound scores were improved after a single day? Or at 4 weeks? Can you please clarify and expand on what the ultrasound score included? I suppose if this is referring to rapid reduction in edema that is plausible. Thank you for clarifying.

61 – consider revising sentence to ‘Although multiple experiences of HILT have been reported, thus far they have been based..’ – removing so far at beginning of sentence

Intro overall – it would improve the introduction to expand on the clinical utility of why increase in surface temperature is clinically relevant – is increase in temperature associated with clinical improvement or are certain temperature increases known to be detrimental and therefore to be avoided? Please expand on the rationale for this study.

82 – was distance to the limb standardized and was ambient temperature controlled or recorded? In the reviewer’s experience, varying distance to the object of interest and varying environmental temperatures may significantly affect temperatures measured thermographically – now I have read farther and see that this is indicated – consider moving this up before the section on treatment

Discussion – as with the introduction, it would be helpful to the reader to include some discussion of the clinical significance of the observed temperature increase – have specific increases in temp been associated with clinical significance or is this an area that requires further research? The authors do not overstate their findings but it would add to the communication if further discussion of context and next steps were added